# Review of Aerial Transportation of Suspended-Cable Payloads with Quadrotors

Julian Estevez [1,*], Gorka Garate [1], Jose Manuel Lopez-Guede [2] and Mikel Larrea [1]

1 Group of Computational Intelligence, Faculty of Engineering of Gipuzkoa, University of the Basque Country (UPV/EHU), 20080 San Sebastian, Spain; gorka.garate@ehu.eus (G.G.); m.larrea@ehu.eus (M.L.)
2 Group of Computational Intelligence, Faculty of Engineering of Vitoria, University of the Basque Country (UPV/EHU), 01006 Vitoria, Spain; jm.lopez@ehu.eus
* Correspondence: julian.estevez@ehu.eus

**Abstract:** Payload transportation and manipulation by rotorcraft drones are receiving a lot of attention from the military, industrial and logistics research areas. The interactions between the UAV and the payload, plus the means of object attachment or manipulation (such as cables or anthropomorphic robotic arms), may be nonlinear, introducing difficulties in the overall system performance. In this paper, we focus on the current state of the art of aerial transportation systems with suspended loads by a single UAV and a team of them and present a review of different dynamic cable models and control systems. We cover the last sixteen years of the existing literature, and we add a discussion for evaluating the main trends in the referenced research works.

**Keywords:** UAVs; suspended payload; collaborative; control engineering; cable modeling

## 1. Introduction

When dealing with the aerial transportation of payloads, the first approach historically taken was to deal with the problem of a load suspended on a crane. In this problem, two types of models could be used: global-mass models and distributed-mass models [1].

Lumped-mass models are characterized by a massless cable, where the payload is lumped with the hook and represented by a point mass. This model is simple while capturing the complex dynamics of payload motion [2–4].

Distributed-mass models are composed of a distributed-mass cable and a lumped point mass modeling the payload. The only model published that falls into this category is the planar model developed by d'Andrea-Novel et al. and Abdel-Rahman et al. [1,5]. However, Choo and Casarella [6], in their review comparing several modeling methods, including some continuous and discrete cable models, arrived at the conclusion that the lumped-mass representation, despite the heavy computer workload needed for its implementation, is the most versatile of them all. This technique models the cable as a finite series of rigid links of lumped masses at the joints.

Focusing on aerial applications, over the past few decades, towed-cable systems have been extensively researched for diverse applications, often promoted by military interests, such as the delivery and retrieval of payloads [7,8], aerostats [9], tethering systems [10,11] and aerial refueling systems [12]. A typical towed-cable system is composed of three components: a towing vehicle, a cable (string or tether) and a towed body (drogue) [13].

As UAV research progressed, the evolution of air transport using UAVs also began to generate interest. This resulted in the possibility of performing a wider range of transport tasks using UAVs [14]. These tasks include various activities, such as transporting larger and diverse objects, examining and maintaining different elements and surfaces and carrying out industrial and emergency-related applications [15]. Most current research is focused on the dynamic modeling and control of the system encompassing the UAVs and the payload. The coupling of the UAV and the payload introduces strong nonlinearities

into established equations that depend on the specific kind of system [16]. Two major carrying strategies have been researched by the scientific community: the direct attachment of the payload to the quadcopter body and the suspension of the cargo with cables [17,18]. For the former, the cargo is attached to the robot body (normally below the center of gravity) through claws, robotic hands or electromagnetic grippers [19]. This method allows for a quicker attachment/release of the load, but it increases the inertia moment of the system, thus making it slower and harder for agile maneuvers and rapid attitude changes [20–22]. For the latter, the load suspended with cables adds degrees of freedom to the inherently underactuated nature of the quadrotor, altering the flight dynamics [23]. In both cases, the system controller needs special requirements added to those used for UAVs without a load [24,25]. However, the fulfillment of the special conditions to obtain fast, stable, rapid and robust flight has no optimal solution, and this leaves the door open for a large number of controller designs.

This survey focuses on the transportation of cable-suspended loads with multirotors, namely, quadrotors. At the moment, recent reviews on the civil applications of UAVs do not cover all the design possibilities, including individual or collaborative schemes, cable modeling or fast maneuvers. Some other research reviews related to UAV payload applications have been published lately. For instance, refs. [26–28] present reviews of multirotors transporting a payload, but they compare different methods for that. Among these last papers, several designs are described: suspended loads, grasping or the usage of arms, and specific issues, such as interactions with objects and the required sensors, are covered. But these papers do not describe the control strategies used in the field. Another recent and highly cited review by Ruggiero et al. [29] pays much attention to quadrotors that have arms and interact with objects, which is one of the main areas of expertise in their research group.

The main contribution of our review is the exclusive study of multirotors carrying suspended loads with cables. The nonlinear nature of drone behavior is further compounded by a greater constraint than when it flies with no load, making it an extremely complex problem and, at the same time, challenging for future applications. We cover the mathematical modeling of cables and control strategies for both individual robots and teams of robots.

It is important to note that, in this research review article, we will not deal with hardware platforms or sensors. Instead, our investigation will be directed toward the mathematical modeling of cables, control strategies and subsequent experimental validation studies (when possible). This deliberate scope allows us to delve deeper into the specific areas of interest, providing a comprehensive analysis and valuable insights while avoiding unnecessary redundancy in the discussion of hardware and sensors, which are often well documented elsewhere in the literature. By narrowing our focus, we aim to provide readers with a more targeted and informative examination of the key facets within the purview of our study.

This article is divided into the following parts: In Section 2, we present the methodology and article selection criteria that we followed in order to complete this review article. Section 3 describes different approaches to cable modeling for the transportation of objects using suspended-load transportation with quadrotors. Section 3 is divided into individual and collaborative transport. Next, Section 4 discusses control strategies for payload transportation by aerial systems, including different optimization strategies. Once again, the section is divided into single and collaborative groups of rotorcraft. The actual knowledge in this field today, future trends in research and technical challenges that still need to be dealt with are discussed in Section 5. Finally, Section 6 gives some remarks about the presented concepts.

## 2. Methodology

The works cited in this article were chosen according to their relevance and interest in the field of modeling and control of the transportation of objects by single or teams of UAVs, more specifically, individual and teams of quadrotors using cable-suspended payloads.

These systems are complex and nonlinear and require elaborate mathematical models to describe their dynamics, as well as to design adequate controllers. Because of the complexity of these systems, the requirement for carrying out practical experiments in addition to simulations was mandatory for the selection made in this survey. Another filter used was the non-inclusion of theses and unpublished dissertations.

In order to evaluate their relevance and interest, the aspects considered were the technical quality of models of both dynamics and control and the innovation of the presented proposal. Finally, in seeking relevant works in the field, the selected articles were manually perused and are presented in a reference list. Table 1 summarizes the criteria used in this survey.

**Table 1.** Article selection search criteria.

| Criteria | Data |
| --- | --- |
| Scientific Database | IEEEXplore, Google Scholar, ISI Web of Knowledge, ScienceDirect |
| Publication Period | From 2007 to November 2023 |
| Keywords | ("quadrotor" OR "rotorcraft" OR "quadcopter" "UAV" OR "multi-rotor" OR "multiple quadrotor" OR "swarm robot" OR "collaborative robots" OR "team of quadrotors") AND ("delivery" OR "transportation" OR "transport" OR "retrieval" OR "cargo" OR "cable" OR "payload" OR "suspended load") |

## 3. Cable Modeling for Payload Transportation with UAVs

In the following paragraphs, different cable models used for suspended-load transportation, both with individual robots and with a team of robots, are presented.

### 3.1. Individual Transport

Early research works about UAVs transporting payloads appeared in the late 1990s [30,31]. For payload transportation by UAVs, the cable treatment is a key factor for modeling the system and the exerted forces experienced by the quadrotor when the lifting, transport and delivery stages have distinct characteristics [32]. During the transport phase, the payload transmits tension through the cable, while in the very beginning of the lifting stage, there is no force transferred through the cable [33]. For simplicity, researchers tend to reduce such force transfer to whether the cable is taut or not [34].

This dynamic model represents the payload as a mass particle and the cable as a massless rigid bar that permanently maintains a constant distance between the payload and the quadrotor and can only transmit axial forces through it. Under these conditions, in 3D scenarios, the payload system is defined by two angles in space, while in planar cases, one angle is enough, similar to a pendulum [35], as can be seen in Figure 1.

Normally, extra restrictions are considered in the dynamic modeling of these systems [33,36,37]:

1. The quadrotor is modeled as a symmetric rigid body.
2. The cable is modeled as inextensible, massless and attached to the center of the quadrotor, and the payload is modeled as a point mass attached to the cable.
3. The mass of the payload is small compared to the mass of the quadrotor, which implies that its motion has little impact on the motion of the quadrotor.
4. The effects of the payload and the cable are treated as an external force applied to the UAV.

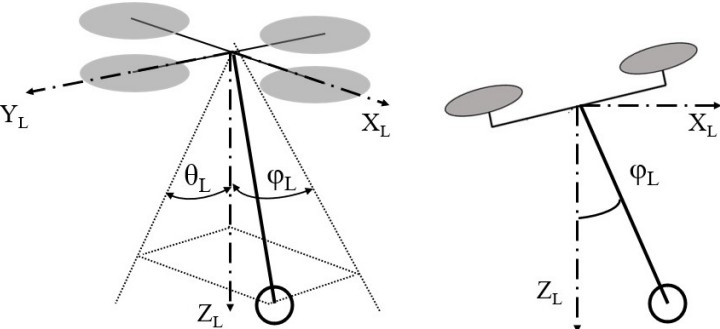

**Figure 1.** Payload with taut-cable modeling in 3D (**left**) and 2D (**right**) scenarios.

Taut-cable approaches became successful because they permitted an easier stabilization of the UAV positioning. Lupashin and D'Andrea [38] proposed a tethered quadrotor and modeled the cable as taut, and they used this cable as a user interaction medium for a low-cost, small hovering UAV in order to stabilize the orientation and the position. With the taut cable, researchers proved that simple inertial measurement sensors are enough for the quadrotor to recover its position and attitude after external perturbations. Following this approach, Sreenath et al. [39] modeled the cable of a suspended load as taut for both the tense-cable and zero-tension cases. In a 3D scenario and a model with nonzero cable tension, the equation dynamics of the system turn out to have 8 degrees of freedom, with 4 degrees underactuated. On the contrary, when the cable transmits no tension, they considered that the UAV and cable form separate systems, and the load is in free fall. In both cases, they validated the models using simulations and real experimentation using taut cables, which permits a high realistic performance for trajectories with curves.

Despite the simplicity of payload modeling, it is a widely accepted technical solution among scientists, as different works from the last two years reveal [40–43], where the mathematical formulation and constraints of the model have remained unchanged.

The limitations of the model of a taut cable are revealed when the quadrotor performs certain critical tasks [17,44,45]. Klausen et al. [46] tested a taut-cable model for aggressive maneuvers and highlighted that there is a substantial load deflection during sudden accelerations and that the payload keeps oscillating when the UAV reaches the hovering state, despite being low-amplitude swings. These limitations are the reason for other cable model proposals.

One of those critical tasks is the lifting of the load from the ground, where the quadrotor and payload system experience different dynamics and a taut cable no longer makes sense. Cruz et al. [47,48] proposed a hybrid model of the cable and UAV, consisting of dividing the process, from lifting the payload to completely separating it from the ground, into three phases (*Setup*, *Pull* and *Raise*), and for each of them, they developed different switching dynamics. These switching dynamics, known as *cable collision* [49], arise when the cable state passes instantaneously from slack to taut; moreover, the UAV undergoes another tension jump when the load is completely in the air. These three steps can be seen in Figure 2.

The *Setup* phase is defined solely by the dynamics of the UAV, while the payload has no effect. At the precise moment the UAV makes the cable vertical, it changes the tension from slack to taut, and it is calculated by the collision effect presented in [49]. Next, in the *Pull* phase, the payload is still in contact with the ground, and thus, researchers take into account the normal force that the ground is exerting on the particle mass; although the cable is still not fully tense, it is already considered taut. Finally, in the *Raise* phase, the cable is tense, and the particle loses contact with the ground.

However, Alothman et al. [50] presented work in which they researched the transition from lifting to transporting the payload, and they split the dynamic equations into two. In the first stage, the quadrotor has no tension at all, and the dynamic equations do not take the cargo into account. In the following phase, when the aerial robot exceeds the height of

the cable length, the system is transformed into a UAV and a slung-load system, modeling the cable as taut. The transition from one phase to the other is more abrupt than in the case of [47], and they validated the model only with simulations. Similarly, Estevez et al. [51] developed a taut cable modeled as a pendulum for the lifting phase, considering no friction with the ground. They did not get rid of any switching dynamics between the phases and validated it through experiments. Moreover, according to the trends in recent years, this payload-lifting procedure remains the simplest possible, and the cable keeps switching from a slack to a taut phase when exceeding a height threshold [52–56].

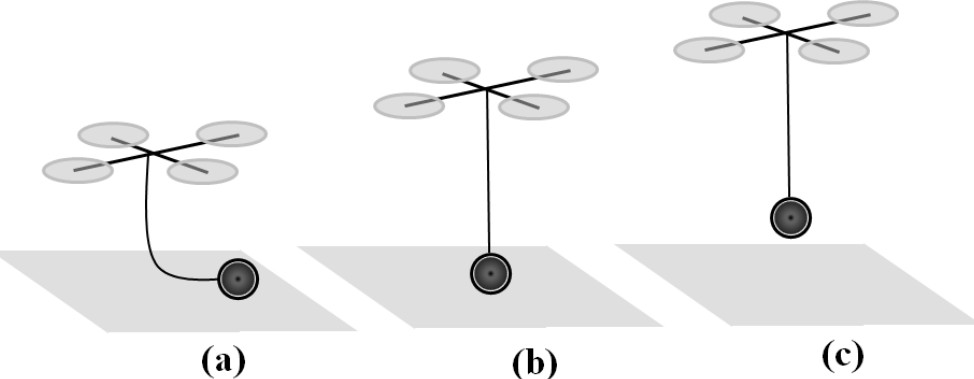

**Figure 2.** The lift maneuver: (**a**) *Setup*, (**b**) *Pull*, and (**c**) *Raise*.

While works seen until now modeled the cable as a massless rigid link, Kotaru et al. [17] considered the cable to be elastic, and they validated their model by focusing on robotic application studies, where cable elasticity cannot be ignored and thus, the rigid-rod model is no longer valid. For that, they included a damping and a spring in their cable model (see Figure 3) and tested the system stability under perturbations and with different damping and stiffness values; however, their study was validated only through simulations. The procedure for modeling this cable as a damper combined with a spring has been applied to the study of other UAV-navigation-related tasks [57], which suggests that the model is still considered valid for capturing the mentioned specific payload effects, particularly large payload swings.

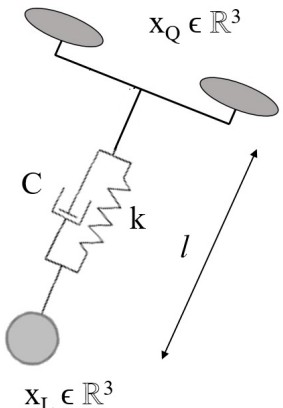

**Figure 3.** Cable model with damping and linear spring. $C$ and $k$ are damping and stiffness coefficients, $l$ refers to the cable length, and $x_Q$ and $x_L$ refer to quadrotor and load positions.

Later, Goodarzi et al. [24] introduced another flexible-cable model, formed by a series of weighted segments of different sizes connected with spherical joints (see Figure 4). The links between joints can elongate, and the researchers aimed to obtain a more precise dynamic model for the aggressive maneuvers of UAVs, with payloads modeled as a serial chain of $n$ connected links, which aimed to prove the stability of a coupled system composed

of a tethered cable and a UAV performing flight maneuvers in 3D. The authors considered the vibrations of the cable through vibrations of the *n* connected links. Nevertheless, to the knowledge of the authors of the current article, this schema has not been widely followed in the literature.

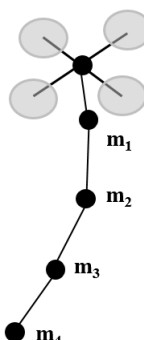

**Figure 4.** Cable model developed by [24].

In terms of landing, the research community has covered the process for UAVs not carrying payloads [58,59]. However, the challenge of landing with a payload has not been studied in depth [32], and the literature reflects just some attempts. One of them was developed by Goodarzi [60], who used a variable-length cable to lower the payload to the ground, and he validated the research through simulations. Next, Qian et al. [32] proved, through real experimentation, that with a precise control design, the assumption of a taut cable is valid for the delivery of a slung payload on the ground.

### 3.2. Collaborative Transport

Collaborative systems are useful for the transportation and orientation of the payload. Actually, the use of multiple UAVs can manage to perform more complex tasks and overcome the limitations of an individual load, such as the enhancement of the load capacity and better control of oscillations [61]. However, these pros come at the cost of increasing the system complexity and aerial vehicle coordination, which must avoid collisions with one another [62]. Their dynamics and additional control requirements are extensively discussed in [29,63,64]. Therefore, the optimization of these variables is not solved yet.

The literature has introduced solutions involving the rigid attachment of multiple quadrotors to objects, as detailed in [65,66]. However, these cases have demonstrated that manipulation is considerably more challenging to achieve than transportation. This difficulty arises from the underactuation property of multirotors [67]. To address these limitations, a reliable alternative has emerged in the form of rigid links connected through spherical joints to the payload, as evidenced in [68–71]. This approach ensures the full actuation of the platform by robots. Furthermore, substituting rigid links with cables enables the design of flexible floating transportation structures. Notably, these cables facilitate the partial decoupling of the vehicle's rotational dynamics from that of the carried payload. Additionally, employing spherical joints enhances the flexibility of robot formation shapes, while the lightweight nature of the cables significantly increases the robots' payload capacity.

Many papers consider the usage of massless rigid links due to their lower complexity [34,64,72]. Michael et al. [73] presented a model to transport a disc-type payload via towed cables with quadrotors. The problem is analogous to that of cable-actuated parallel manipulators operating in three dimensions, as both types of manipulators are designed to control the pose of the payload with varying robot positions and in aerial systems, and the payload orientation is modified with aerial cable towing performed by quadrotors (see Figure 5). They used three different approaches to solve the problem: inverse kinematics of the payload and UAVs, the direct problem and an optimization of that last direct problem. Moreover, the same research group modeled the transportation of a suspended DLO (deformable linear object) [74] and designed a deep mathematical background for the

formulation of the system's stability and control, but with severe dynamic limitations and quasistatic conditions.

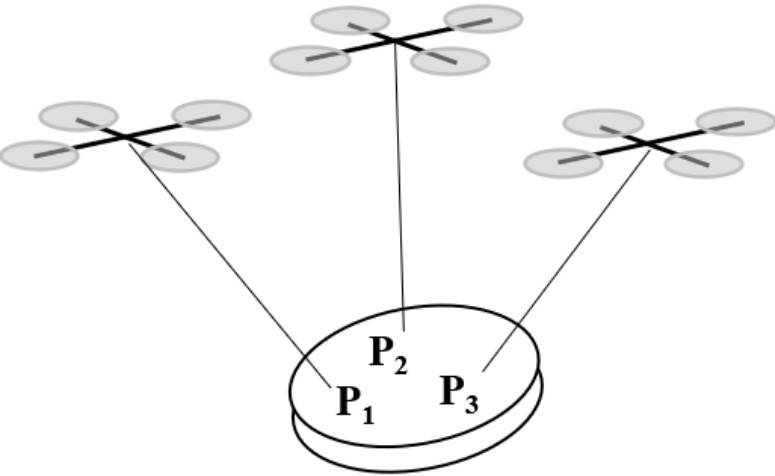

**Figure 5.** A team of three point-model robots manipulates a payload in three dimensions [73].

Another cable alternative was proposed by Pizetta et al. [62], who used a model of elastic cables, and they relied on their traction force to reject the disturbances created by the payload on the UAV. Moreover, they proved that their proposal is valid for lifting the payload from the ground, using a single system of dynamics and a series of assumptions: (1) the cable is massless; (2) when the load is on the ground and the cable is slack, there is no effect on the vehicles; (3) the aerodynamic effects on the load and vehicle are negligible. They simulated their system with two UAVs considering only the longitudinal plane for payload swings.

The group transportation and orientation of a rigid two-dimensional payload is achieved by using a cable system that is modeled as a series of connected links [75] (see Figure 6), as demonstrated in the works presented in the previous sections, [24,76], which demonstrate the robustness of this approach to payload transportation with UAVs.

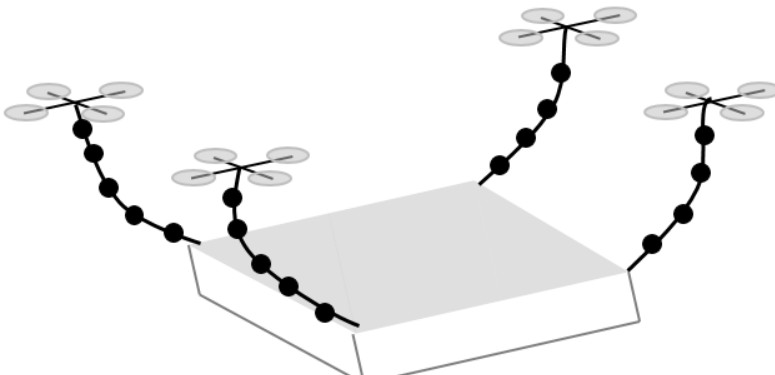

**Figure 6.** Quadrotor UAVs with a rigid-body payload. Cables are modeled as a serial connection of an arbitrary number of links.

However, taut cables are not the only mechanical elements that have been researched in order to achieve a better representation of reality, and [77] developed what they call *tensegrity muscles*. This element consists of a finite number of tensegrity prism cells, where each prism cell is made of tethers and rigid bars, and they can apply tension and compression efforts. The key to their approach is to consider these elements as a continuum deformable element, which permits them to scale the number of UAVs in the aerial transport task, thus gaining some advantages, such as an increase in robustness, reconfiguration

capabilities and the ability of the system to navigate in constrained environments, such as narrow channels.

Catenaries are widely accepted by the scientific community as a dynamic cable model representation and have been used in submarine mooring cable simulations [78–80]. However, models based on catenaries have not been fully exploited for aerial towing, despite the promising results presented in previous works [78,81,82]. Works using this cable formulation defend the advantages of catenaries for their quasistatic configuration, ease of computation and well-established rigid–solid mechanics equation. Catenaries can be used to formulate discrete or continuous models. Estevez et al. [83] presented a collaborative quadrotor system for DLO transportation using catenaries, as shown in Figure 7. They created an equiload quadrotor height configuration for the transportation of a cable with the same vertical load for each robot under the following assumptions:

- The cable diameter is negligible compared to its length. Thus, the cable can be modeled as a 1D object.
- The mass per unit length of the cable is constant.
- The cable cannot elastically lengthen (Young's modulus is large).
- There is no torsion in the cable.

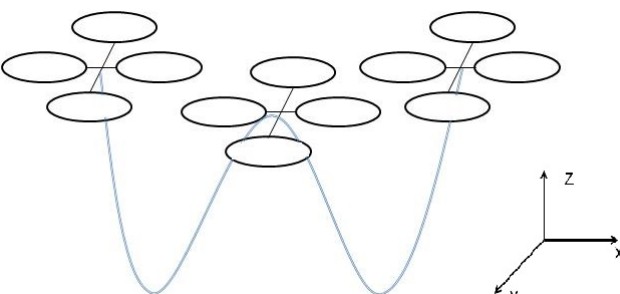

**Figure 7.** Transportation of DLO with a team of quadrotors.

The same research group evolved their work and turned their proposal into a hybrid parabola and catenary switching model, so the team of UAVs does not collapse the catenaries when they get too close in aggressive maneuvers or sharp trajectories [84]. The approach to modeling cables with catenaries has been adopted by other research groups, such as [85,86], who experimentally tested the validity of the solution. However, both of them just use two rotorcraft in the formation.

However, very few of the multi-robot transport proposals in this section take into account lift-off or landing procedures. Although Goodarzi and Lee [87] and Michael et al. [73] performed experiments showing that their systems are able to lift and land, they do not describe the mathematical modeling of the cable for a transient regime. Including the dynamic properties of the cable during these phases increases the cost of computation.

The scientific literature in this field is very scarce and relies on some simplifications to ease the calculus and computational cost [88]. For instance, Bacelar et al. [89] presented two AR Drone 2.0 quadrotors for transporting a suspended load and specified that they are always assumed to be taut. Next, Pizetta et al. [90,91] proposed a system of two quadrotors transporting a point-mass load with two cables formed by point-like masses joined by springs and dampers (Kelvin–Voigt models). This cable model is able to absorb the contact with the ground and linearly reduce the tension of the cables. Geng et al. [92] and Goodman et al. [93] used the same cable dynamic model. Lately, some other researchers have reproduced this schema for loading bar- or rod-shaped payloads, as can be seen in Figure 8. While elastic cables offer the advantage of mitigating impulsive forces on the bar, excessive oscillations can induce unwanted forceful movements, potentially jeopardizing the safety of the collaborative task. Therefore, the implementation of elastic cables with enhanced stiffness and damping is crucial to safeguard the bar during the transportation

process. For instance, Goodman et al. again validated their results with simulations in [94], while Gabellieri et al. [95] validated their proposal with real experiments.

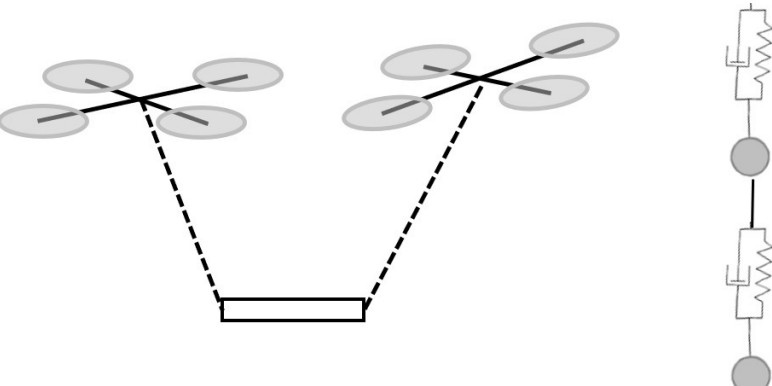

**Figure 8.** Quadrotors transporting a cable-suspended bar (**left**) and cable section represented by mass–spring–damper systems (**right**).

Finally, Shirani et al. [88] presented a mathematically simple cable collapse-and-collision model for lift-off and landing based on the geometric coordinates and distance between the UAVs and the payload. Surprisingly, the trend in recent years has been to extend the cable model that switches the state of the cable from slack to taut when a height threshold is achieved with a team of rotorcraft [69,91,96,97], which is apparently the simplest formulation, mathematically speaking. This slung-load option prevails as one of the most used alternatives, both for experiments including payload lifting and for those without it. However, for the former, these studies are limited to smooth maneuvers.

## 4. Control Strategies

Dynamic models of multirotor-type UAVs usually consider that the geometry and mass distribution are symmetrical, which allows for some simplifications of the dynamic equations. The mass distribution of an aerial robot with a payload is no longer symmetrical, as it varies widely with the movement of the manipulator or cable mass. In the following paragraphs, we discuss the state of the art in control design for several of the above configurations.

### 4.1. Individual Transport

During flight, the payload adds passive dynamic effects, which could come from either the cable or the payload, and generates swinging that modifies the dynamics of the UAV and alters its dynamic performance. The most common way to deal with the perturbations caused by the payload is to stabilize the system, minimizing the load swings [36,98]. Alternatively, some works used feedback control to track the desired load trajectories or trajectory-planning algorithms for the quadrotor-with-load multi-body system [99]. Some other researchers divided the cable model into different subsystems. In these hybrid dynamic models, each subsystem has its specific and specialized controller, and the switching among the simple controllers is performed by a supervisory system [39,47]. In contrast, other groups worked on the search for a general solution to the problem of suspended-cable transport, no matter what stage the mission is in [23,50,100].

The dynamic system of a quadrotor with a hanging payload in 3D has eight degrees of freedom and only four control inputs. Thus, the same four model simplifications mentioned in Section 3.1 referring to the UAV and payload are assumed, but still, the four degrees of underactuation make the controller design challenging [101]. This subsection is divided into three different control-oriented approaches for the UAV and the payload system. They include swing attenuation of the payload, optimal trajectory tracking and control for aggressive maneuvers.

#### 4.1.1. Anti-Swing Control

The control solutions for UAVs transporting slung-load-type payloads (see Figure 1) are derived from crane control models, which is especially useful in cases of a free-swinging load. In particular, this control strategy tries to damp excessive payload oscillations that would greatly alter the UAV dynamics along the target trajectories [46].

Palunko et al. [102] developed a dynamic programming approach for a coupled system of a UAV and cable for the control of a swing-free trajectory and validated it using an AscTec Hummingbird. However, the dynamic uncertainties in these scenarios tend to deteriorate the performance of open-loop configurations like this. Therefore, researchers opt for different closed-loop control alternatives to handle the quadrotor motion and the payload swing. For instance, de Angelis et al. [103] proposed a two-time-scale controller that can deal with the trajectory and position tracking of the payload and the UAV simultaneously.

On the other hand, Nicotra et al. [104] describe a nested saturation control method for the transportation of a slung load with a quadrotor and validated the method for the rejection of external disturbances and wind during flight. Guerrero et al. [25] used a Hamiltonian approach to control the nonlinear system, wherein the interconnection and damping assignment-passivity-based control scheme is utilized to stabilize the swinging oscillations. Notably, their method exhibits independence from the swing angle, effectively suppressing swing motions along the designated flight trajectory. However, certain constraints are inherent in their methodology. These include assumptions such as the cable being both stiff and massless, the payload being considered a mere mass point and the system being analyzed within the longitudinal plane. Xu et al. employed both PD and sliding-mode control for the quadrotor and slung-load system, demonstrating that sliding-mode control offers superior robustness to the effects of the cable-suspended load [105].

In reference [106], a straightforward active-model-based control system was developed for the quadrotor slung-load setup, augmented with a Kalman filter (KF) to bolster performance by mitigating disturbances. In experiments, the widely adopted Pixhawk controller was utilized. The findings demonstrate the enhancements achieved with the active-model control system, as evidenced by a comparison of the system performance with and without the proposed controller. Another successful anti-swing strategy consists of an adaptive controller, which different researchers developed in order to make control proposals for UAVs with some unknown variables, such as the payload mass [107,108]. The cited adaptive controller allows the online estimation of the unknown parameter.

Liang et al. [109] proposed a minimum-time trajectory-planning method, designed specifically for unmanned quadrotor transportation systems, which presents some advantages over existing methods by simultaneously taking into full consideration the system's nonlinear dynamics and various constraints.

One effective feed-forward method is the input-shaping theory, which has proven to be a practical and effective approach for reducing vibrations [110,111] and has also been applied in various works researching UAV and payload systems. Huo et al. [112] present a double closed-loop control strategy combining PID position control and an input-shaping method, which makes the quadrotor follow the desired trajectory and eliminate the vibration of the load simultaneously. However, this vibration reduction strategy can also be combined with other control methods. Kuznir et al. [113] included input shaping in a sliding-mode-based controller, and Slabber et al. [114] used it along with an LQR controller.

#### 4.1.2. Optimal Trajectory Following

T. Lee's geometrical controller [115,116] enables the tracking of the desired path for both the payload position and attitude. To achieve this, the Voronoi tessellation technique is employed to perform trajectory planning while also considering the constraints related to collision avoidance. On the other hand, in their work, Sreenath et al. [39] pursued the study of how to control the position of the load when the load undergoes large swings, reaching zero-tension finite moments. They designed a feedback controller and proved that the system is differentially flat [117] and valid for the control of the UAV and payload

system. Considering external disturbances, Qian and Liu [118] carried out payload transportation with a UAV under windy conditions. They implemented a cascaded closed-loop control configuration, where, for the outer loop, a UDE-based (uncertainty and disturbance estimator) translational control law is proposed. This loop is responsible for stabilizing the quadrotor along a predetermined trajectory and estimating the aggregate disturbances using a low-pass filter. The inner loop comprises an attitude-tracking controller, which adjusts the direction of the lift vector to align with the thrust vector direction and can asymptotically follow the reference force generated by the outer-loop.

Next, Cabecinhas et al. [119] developed a quadrotor with a slung-load model subjected to uncertain disturbance parameters and a payload, where the objective was to design a dynamic feedback controller that makes the slung-load follow a large class of embedded curves and leads the system smoothly along the given trajectory. Similarly, Tagliabue et al. [70] presented research work based on robust control theory to guarantee stability in worst-case scenarios, considering several physical uncertainties of the system. There are not many research works in this field that use machine learning yet. The first attempt was made by Faust et al. [120], who trained a model-free reinforcement learning (RL) algorithm for the transport of a payload with a UAV, where the reward goes toward the minimization of the residual oscillations of the payload. The dynamics of the model are not necessary in this model-free technique, and the work proves that, once trained, the algorithm is able to create multiple paths. The RL controllers exhibit a notable reduction in payload swings during dynamic maneuvers along the desired trajectory, affirming the effectiveness of this approach. Similarly, Li et al. went a step forward and developed the same strategy with a deep reinforcement learning approach [121].

Over the past few years, researchers have been increasingly drawn to the Model Predictive Controller (MPC) as a means to attain optimal trajectories during payload transportation [122]; an MPC is a control strategy well suited for designing a distributed and multi-level control system, where the reference trajectories for UAVs are computed as a solution to a constrained optimization problem. For instance, Lee et al. [123] combined an MPC with a Proportional Derivative (PD) controller using the swinging-load-angle error as an input and correcting the translational controller along a planned trajectory. Estevez et al. [44] proved that this control strategy is still valid for a payload modeled as a double pendulum. In [124], they used an MPC with Sequential Linear Quadratic (SLQ) control in a scenario where sudden obstacles can appear, and the experiments show that the aerial system is able to fly in these dynamic scenarios. However, the MPC turns out to be a more valid tool when considering aggressive maneuvers, as will be described next.

### 4.1.3. Aggressive Maneuvers

The aerospace industry has been highly prioritizing the agile flight of aerial vehicles for almost 80 years [125]. Similarly, urgent situations have emphasized the requirement for agile autonomous systems that can perform search-and-rescue operations promptly [126]. In order to enable such opportunities, UAVs must precisely monitor agile trajectories under model uncertainties, such as unknown drag coefficients, and external disturbances, such as varying payloads or wind gusts.

The authors of [127] successfully demonstrate the accurate tracking of aggressive quadrotor trajectories by utilizing a cascaded geometric controller with Incremental Nonlinear Dynamic Inversion (INDI). Sreenath et al. [128] built a geometric controller for the position tracking of a quadrotor that handles a payload with large swings, and thus, it can be considered an aggressive maneuver for short time periods. However, as rightly noted in [129,130], INDI and traditional geometric controllers are entirely reactive and lack the capability to plan over a prediction horizon. To cope with this and attain advanced tracking performance at speeds up to 14 m/s, Nonlinear Model Predictive Control (NMPC)-exploited data-driven techniques to enhance the model's fidelity was developed in [129]. Due to the computational intensiveness of the learned model, control commands were calculated off-board the quadrotor and sent through wireless communication. However,

the model cannot adjust to online parametric changes, such as payloads or a decrease in actuator efficacy, because the model parameters are obtained offline. However, in the model developed by Hanover et al. [125], they couple an $L_1$ adaptive control law with NMPC, and thus, thanks to the adaptive control law, the control system can learn modeling uncertainties online and immediately compensate for them.

Due to the high computational cost of NMPC methods [131], we could think about the usage of alternatives, such as Linear MPC (LMPC). However, position control [132] and motion control based on small-angle assumptions [133] are the sole focus of many studies that employ LMPCs. As a result, LMPC techniques are inadequate at capturing the nonlinearities present in rotational dynamics, leading to inferior performance compared to NMPC methods [134].

*4.2. Collaborative Transport Control*

The most popular methods for the transportation of payloads with a team of rotorcraft are the usage of cables, robotic arms and grasping. Each alternative has its own advantages and disadvantages, but overall, cable transportation is preferred due to the low cost and the lack of complexity [72]. The main disadvantage of cables is the load instability, particularly in hazardous environments. In contrast, robotic arms and have the advantage of not requiring a cable connection, but they are limited by high cost and energy consumption [29], while grasping's drawbacks consist of the contact materials and the fact that the vibrations of the payload tend to be transmitted to the UAVs.

To address the challenge of the collaborative manipulation and transportation of a shared object using quadrotors and cables, Maza et al. [135] created a PID controller. However, the suggested controller technique lacks the ability to diminish oscillations in the underdamped system. Bacelar et al. devised a controller method for collaborative load transportation using two aerial vehicles [89], which combine a Linear Quadratic Regulator (LQR) controller with a Kalman filter. The approach was experimentally tested using two commercial quadrotors equipped with ultrasound height, IMU and frontal cameras.

Ariyibi and Tekinalp [136] introduced a hierarchical control system for two quadrotors transporting a slung payload. The system employs a linear controller for the position loop and a quaternion-based nonlinear controller for the attitude loop. To verify their controllers, the researchers used a simulation code to test rigid and flexible loads. Klausen et al. [137] created a collaborative algorithm resistant to environmental disturbances, specifically for unknown masses. The proposed control algorithm is decentralized, with each quadrotor using the relative velocity and position of its neighbors to achieve the desired formation shapes. In [138], a decentralized Lyapunov-based controller is proposed for the aerial manipulation of a cable-suspended payload using two aerial robots. The objective of the work is to regulate the position and attitude of the payload without requiring the robots to communicate through a master–slave admittance controller. In [139], a distributed approach to collaborative transportation using an MPC is demonstrated.

The work by Gimenez et al. [140] introduces a new kinematic formation controller that employs null-space theory to enable a payload hanging from two rotorcraft UAVs via flexible cables to follow the desired trajectory. The proposed controller accounts for both wind disturbance and obstacle avoidance. A later work by the same authors, validated the landing stage of two rotorcraft transporting a rigid payload through experiments with the same control technique [141].

Finally, Lee et al. [142] extended their geometric controller to multiple quadrotors that were used for load transportation with a team of robots. However, these methods assumed that the payload is a point mass, which is impractical since the actual load does not have any neglectable geometric dimensions and can have the same size as a quadrotor. To overcome this limitation, a geometric nonlinear controller was designed in [116] for the group transportation of a rigid-body payload. However, the method requires knowledge of the links' orientations and angular velocities, which are difficult to compute in practical scenarios. Therefore, Sharma et al. [143] devised a dynamic model of the system where

they consider the coupling between the payload and the rotorcraft, and thus, they created a quad-centric approach that avoids the mentioned difficult-to-obtain parameters. The previous works mentioned in this paragraph neglect the coupling.

## 5. Discussion

This section presents an analysis of the performance of the cable modeling and control systems discussed thus far, both for individual and for collaborative transport, and highlights some significant observations.

First, we show the main cable model and control system alternatives and their pros and cons according to the consulted literature. In Table 2, the main highlights of cable models are collected.

**Table 2.** Highlights of cable models.

| Single- and Multi-UAV Transportation Systems | |
|---|---|
| **Model** | **Description** |
| Taut cable | It is the most common and used model for both single and collaborative UAV systems. Mathematically simple, the payload is represented by a mass particle, and the cable is represented by a massless rigid bar that permanently maintains a constant distance between the payload and the quadrotor. It represents most of the cable dynamics with enough detail. It is al used for loose cables and for lifting payloads from the ground. Cable nonlinearities in aggressive maneuvers are not well represented. Particularly in multi-UAV systems, more real experimentation is required.<br>References: [39,40,46,48,50,51,53,89,144] |
| Flexible cable | Flexible cables enhance some dynamic properties in simulations. The most used models are the spring-and-damping model and the cable formed by a series of weighted extensible segments of different sizes, connected with spherical joints. They are a better alternative for both performing aggressive maneuvers and lifting objects from the ground than taut cables.<br>References: [17,24,60,93,94,141,145] |
| Catenaries and tensegrity muscles | There is scarce research in the literature on these models, and they are designed for very specific tasks. Catenaries are used as the dynamic model for cooperative cable transportation. Tensegrity muscles are modeled for representing tension and compression efforts with tethers and rigid bars in multi-UAV systems too.<br>References: [77,83,146] |

Next, a summary table of control systems is shown in Table 3:

**Table 3.** Highlights of control systems.

| Single and Multi-UAV Transportation Systems | |
|---|---|
| **Model** | **Description** |
| PID and controller gains | Despite being traditional, it is one of the most used control systems in single- and multi-UAV transportation systems. It is usually based on a closed-loop cascade control circuit. Some authors even use PD controllers and apply it to both anti-swing and path-following strategies. In collaborative systems, there is no a clear trend between centralized and decentralized systems. However, according to some authors, this control system does not cope with aggressive maneuvers. Again, in collaborative UAV systems, more real experimentation is needed.<br>References: [39,105,106,109,140,147] |
| MPC and geometric control | These nonlinear control systems are the most used methods for aggressive maneuvers and other transportation tasks, particularly in multi-UAV transportation systems. MPC can be NMPC, and it can be combined with other control systems, such as LQR or PID. Although simulation reflects the dynamic effects with enough detail, more real experimentation in collaborative UAV systems is needed.<br>References: [44,124,125,129,132,139] |
| Deep learning | There is still scarce research based on these tools, but it might widen the spectrum of possibilities for the enhancing the autonomy and open-ended tasks in single- and multi-UAV transportation systems.<br>References: [120,121] |

*5.1. Trends in Technology*

Upon reviewing all the cited literature and assessing the range of applications achieved, it is evident that in the last several years, there has been significant progress in load transportation via quadrotors using cables. This progress can be attributed to advancements in modeling and improvements in path-planning algorithms, which, when coupled with nonlinear controllers, have enabled aerial vehicles to both transport a swing-free payload and execute aggressive maneuvers. Notably, the use of hybrid modeling to address changing scenario restrictions, along with differential flatness to generate smooth and feasible trajectories far from the near-hover-equilibrium point, decreased the failure rate and bolstered the control robustness that was lacking in individual and collaborative quadrotor-with-load systems. Furthermore, the analyzed works have demonstrated the necessity of employing attitude controllers within a hierarchical control framework to manipulate the pitch, roll, yaw and motor speeds and react to passive dynamics and disturbances arising from the load while in motion.

The aforementioned advancements have been extended to group robotic systems, enabling the completion of tasks beyond the capabilities of a single robot. The collaborative transportation of heavy loads, the orientation of large loads and the use of multiple robots to carry individual loads in dynamic environments are now achievable. This breakthrough is a game changer, as it permits robots to execute tasks in real-world scenarios rather than being limited to laboratory-restricted conditions.

In the following paragraphs, the authors will highlight the main trends in some key aspects of the research of quadrotor and payload systems in light of the vast number of analyzed research articles. All the revised works relied on simulation experiments, which is revealed to be the most common validation method. Nevertheless, real indoor experiments are increasingly frequent, both in cable modeling and in control system proposals. Some relevant cited works in this article that confirm this trend are [38,46,70,74,89,109]. However, researchers have encountered the difficulty of measuring some variables, such as the position of the payload or the swing angle [98], even in indoor experiments. Mainly for this reason, there is very scarce research using outdoor experiments. We consider that it is critical to contrast simulation results with indoor experimentation, and it would be advisable to extend them to outdoor settings. There is still a long way to go, as it is really challenging to measure UAV position and speed outdoors.

In the beginning of the analyzed literature, research works used a single quadrotor. However, as technology advanced, the scientific community paid greater attention to team quadrotor systems and the communications between different robots, although this point is out of the scope of the current article. To our knowledge, the taut-cable model remains the most common cable model for both individual and cooperative transport, as many references prove [38,46,69,91,97]. Nevertheless, there is a wider spectrum of possibilities in control system proposals, and we suggest that the scientific community explore learning-based systems to test their robustness for transportation tasks with UAVs.

Another key aspect of the research of suspended-payload and quadrotor systems is the presence of disturbances. As mentioned previously, there is scarce research using outdoor experimentation, and thus, real wind disturbances are almost non-existent, and wind just appears in simulation works, such as [40,69,118]. When validation is indoors, wind is proven not to be the most relevant disturbance, but most researchers work with unknown payload masses or variations in the planned path. Some examples of these disturbances are in [107,114,144].

Finally, we consider that the coupled and decoupled treatment of the payload and UAV system is a relevant decision to the field. Although there is a large number of research articles that consider the coupled system of two elements of the system, we observe that the trend is to implement more decoupled dynamic models, which are easier to handle. Some of the observed articles demonstrating this are [40,44,118,142]. According to validation studies, the results of decoupled treatment are as good as the results of coupled systems.

However, in the case of payloads attached to the body of the quadrotor, this assumption may not be true.

*5.2. Future Challenges*

In the presented real-world experiments, methods with low computational cost have usually been prioritized over optimality criteria, leaving nonlinear and hybrid path optimization still not fully researched. This problem solving becomes more necessary for dealing with uncertain scenarios in highly unstructured dynamic environments, where tasks might require the proper coordination between UAVs, and some dynamic parameters are hard to know in real time.

The presence of external disturbances to the suspended load can impair the performance the controllers in both individual and collaborative transport. Thus, new directions for controlling the quadrotor with a suspended load, which will involve advanced control theory techniques, such as adaptive dynamic programming and deep learning control, are a future challenge. Moreover, there is a lack of real experimentation that sets limits on unrestricted flight simulations, as reported in the previous subsection. There seems to be an unaddressed research gap concerning test platforms tailored for vehicles carrying suspended loads. Consequently, the authors assert the importance of prioritizing the development of such testing platforms for vehicles with suspended loads.

In addition to that, as far as the authors are aware, the literature on the cooperative transport of suspended payloads is still one step behind individual transport. Thus, efforts should be dedicated to closing this gap and further developing mathematical models and experiments.

There is limited availability of published data, especially for outdoor applications. It is important to keep in mind that autonomous flight with a suspended load aims to serve outdoor applications, and therefore, this should be the focus of future efforts of researchers in this field.

## 6. Concluding Remarks

Through the works cited and discussed in this survey, remarkable advances in cable-suspended systems are demonstrated. The nature of each specific task to complete will determine the most suitable control approach, whether it be cable modeling or parameter balance. Many solutions to a broad range of problems have been reported and discussed in essence, although not exhaustively. A more in-depth study and extensive analysis would be necessary to provide more detailed descriptions. However, since the main aspects of load transportation with quadrotors using cables have been identified and presented, we consider that this document will be a solid basis and starting point for researchers to keep advancing in these fields and to introduce the state of the art to newcomers.

**Author Contributions:** All authors contributed to the study conception and design. Material preparation, data collection and analysis were performed by J.E., G.G., J.M.L.-G. and M.L. The first draft of the manuscript was written by J.E., and all authors commented on previous versions of the manuscript. All authors have read and agreed to the published version of the manuscript.

**Funding:** The work in this paper has been partially supported by FEDER funds for the MICIN project PID2020-116346GB-I00, research funds from the Basque Government as the Grupo de Inteligencia Computacional, Universidad del Pais Vasco, UPV/EHU, with code IT1689-22. Additionally, the authors participate in Elkartek projects KK-2022/00051 and KK-2021/00070. The authors have also received support from Fundacion Vitoria-Gasteiz Araba Mobility Lab.

**Conflicts of Interest:** The authors declare that they have no known competing financial interests or personal relationships that could have appeared to influence the work reported in this paper.

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
