# Peer review of "Review of Aerial Transportation of Suspended-Cable Payloads with Quadrotors"

_drones, doi:10.3390/drones8020035_

Round 1

Reviewer 1 Report

Comments and Suggestions for Authors

General comments:

This paper presents a survey of research into the transportation of loads by suspended cables using quadrotor vehicles. The section presented is very relevant in this area and the references cited, for the most part, are directly aligned with the proposal of the work.

Mandatory consideration:

Given that the work is a historical analysis of other works in the literature, I understand that the aim is to present a conclusion regarding the review carried out. In this case, the main contribution of the work would be a table showing the authors' opinions on the subject, in order to guide readers/researchers on the subject. In other words, create an evaluation table of all the works cited, scoring/weighting them:

  • Type of validation: simulation, indoor experiment, outdoor experiment.

  • Individual or cooperative navigation

  • With or without external disturbances. With or without modeled disturbance.

  • How the load is treated: disturbance or part of the model,

  • Other parameters or approaches you deem relevant.

From there, based on these evaluation parameters, I will point out the most relevant and promising articles from the authors' point of view.

Important considerations:

1- There are works in the literature considering the transportation of bar-shaped (rod-shaped) loads with two drones. I strongly suggest that you search for these papers and replace the mention of application with helicopters, in order to maintain the scope of the manuscript. It is worth noting that the transportation has been experimentally validated.

2- Given that Tables 2 and 3 do not provide any new information in relation to the text presented in Section 4, in my opinion, they can be eliminated without losing the content of the work.

Consideration for improving the work:

1- Eliminate Section 3 and only mention that the task of transporting cargo by a UAV has a modeling similar to that of a crane. In this new paragraph, cite the references already cited.

2- Improve the quality of Figure 2 by including the image of a UAV. Insert the reference axes to make it easier to understand, describe what each angle is.

Minor corrections:

1- In Table 1, the suspended load is repeated.

2- Reword the text between lines 178 and 180.

3- Replace "an UAV" with "a UAV".

4- Check all the references, as some are missing data.

Comments on the Quality of English Language

The manuscript is well-written and needs minor adjustments (corrections) in terms of grammar and typing.

Author Response

Dear reviewer, please find the reply attached to this message.

Reviewer 2 Report

Comments and Suggestions for Authors

The paper is well organised to adequately summarise the study of aerial transportation of suspended-cable payloads with quadrotors.

There is a minor concern about the trends in technology regarding the learning-based methods to implement this kind of system.

Author Response

Dear reviewer, please find the reply attached to this message

Reviewer 3 Report

Comments and Suggestions for Authors

Dear Editor.

Thank you for inviting me to review the paper ‘Comparative study of aerial transportation of suspended-cable payloads with quadrotors’.

TYPE OF SCIENTIFIC PAPER

The Abstract states ‘In this paper we focus on the current state of art of aerial transportation with suspended loads by means of single or a team of UAVs, and present a review of different dynamic cable modelings and control strategies’.

To define state of the art is a significant ambition. Based on this, I wonder what type of scientific article this paper aims to be. The authors use terms as review and survey. The method in the paper does not satisfy systematic review, scoping review or other definitions of a review paper.

METHOD

The authors present keywords and sources for their literature research, but they do not inform how many papers were found, how they selected the included papers, how they excluded papers, and the reasons why findings were excluded. I found that application of the presented keywords as stated in table 1, gives quite many hits in several scientific sources. By simply using Google Scholar, one of the sources presented by the authors, I got 33200 hits in the period 2007-2023 defined by the authors, and even 8000 hits after 2022. Of course, this contains a lot of insignificant sources and varying quality, but it illustrates a major challenge that must be addressed and explained to the reader. The background for the included articles and how they represent state of the art covering the last 16 years of ‘existing bibliography’, must be clearly defined.

The title states ‘a comparative study of aerial transportation or suspended cable payloads with quadrocopters’.

It is unclear to me how this ‘comparative approach’ is applied and what the outcome is. There is a presentation of 148 cited papers, but the extent of the presentation from the differing publications varies a lot across the papers cited. Some just briefly, others in some detail. A clear systematic analysis is not obvious to me. The tables 2 and 3, summing up the listed papers, does not present a clear strategy for the synthesis of a ‘comparative study’. It may very well be my fault, but I did not catch the ‘comparative dimension’ in this setup. May be a column stating pro et cons could have enriched the presenting tables. As it is, I observe the major body of the text as a listing of different papers of topics. The intention may be good, but it is completely left to the reader what's the main pros and cons and value of the each paper and this articles contribution.

There is a somewhat confusing terminology related to the aircrafts. It is very well known in the literature that there are several synonyms for drones/UAVs/multicopters. The authors use UAV, quadrocopter, robot body in a mixture that cause challenges for the reader. Is the use of these interchanging terms reflecting differing physical structures, or is it just a varying terminology/names for the same structure?  In the latter case, why use differing terms?

As mentioned previously, I do not find the comparative dimension very obvious, and the multiple topics presented is not clearly summarized into an overall conceptual conclusion. There should be a more clearly defined ‘RESULTS’ section where the major findings, as interpreted by the authors. are portrayed.

DISCUSSION

The discussion is rather brief, and as I see it, does not compensate for the above missing thematic.

In the statement ‘the bibliography of cooperative transport of suspended payloads is still one step behind individual transport’, I assume ‘individual’ is related UAVs operating single-handedly? Or other?

I believe that the paper may be improved by a more limited focus and a more detailed presentation and discussion related to the ‘comparative’ perspective as announced in the title. Filling a potential gap in mathematical modelings and experiments could be discussed in the perspective of what are the major challenges in obtaining such models for real life, outdoor applications. Also, a discussion of this topic related to the extremely fast development of unmanned aircrafts with very heavy cargo capacities may possibly influence the long-run need for ‘team’ UAVs?

I am sorry to say that I did not find this paper as a solid base and starting point to introduce the state of the art to newcomers.

The paper needs English editing as there are multiple misspellings and statements that are not easy to understand.

Comments on the Quality of English Language

The paper needs English editing as there are multiple misspellings and statements that are not easy to understand.

Author Response

(The authors gave the same response as above.)

Reviewer 4 Report

Comments and Suggestions for Authors

The paper explores literature to get the state of the art when it comes to aerial cranes and cooperative transportation of suspended load by unmanned aerial vehicles.

There is a class of aerial cranes that this paper ignores. These are the feedforward based methods which use methods from vibration mitigation such as Input shaping. I will suggest that the authors take a look at this class of controllers. Even motion planning with vibration mitigation, modelling with non-Newton based methods (such as Lagrangian etc) are not mentioned.

I understand the statement by the authors that most experimental implementations are not in the outdoor settings but the question is what does this caveat mean and  how does that diminish or strength the listed studies?

On page 2 line 36 - 38, what does this statement mean?

How does the section on the study of cranes tie into the discussion. I think the authors can improve this section to fit the overall aim of the paper 

Comments on the Quality of English Language

The grammar should be improved

Author Response

Dear reviewer, please find the reply to your comments attached to this message.

Round 2

Reviewer 1 Report

Comments and Suggestions for Authors

The revision has addressed practically all the suggestions made. The new version is really an improvement on the previous one.

However, given that the article is a comparative study, I would still suggest a table in which the authors point out trends in the field. This would be of great help to new researchers entering the field.

Comments on the Quality of English Language

In my view, the article is well-written and requires minor revisions to the English language.

Author Response

Dear reviewer,

Thanks a lot for your consideration on our effort of improving the article. As you suggest, we include in the new version of the revised article two summary tables in Discussion section where highlights, pros and cons of the cable models and control systems are shown.